

# Functional verification of the *JmLFY* gene associated with the flowering of *Juglans mandshurica* Maxim.

Jiayou Cai[1,2], Ruoxue Jia[1], Ying Jiang[1], Jingqi Fu[1,2], Tianyi Dong[1,2], Jifeng Deng[1,2] and Lijie Zhang[1,2]

[1] Shenyang Agricultural University, Shenyang, Liaoning, China
[2] Key Laboratory of Forest Tree Genetics, Breeding and Cultivation of Liaoning Province, Shenyang, Liaoning, China

## ABSTRACT

In this study, a pBI121-*JmLFY* plant expression vector was constructed on the basis of obtaining the full-length sequence of the *JmLFY* gene from *Juglans mandshurica*, which was then used for genetic transformation *via Agrobacterium* inflorescence infection using wild-type *Arabidopsis thaliana* and *lfy* mutants as transgenic receptors. Seeds of positive *A. thaliana* plants with high expression of *JmLFY* were collected and sowed till the homozygous T3 regeneration plants were obtained. Then the expression of flowering-related genes (*AtAP1*, *AtSOC1*, *AtFT* and *AtPI*) in T3 generation plants were analyzed and the results showed that *JmLFY* gene overexpression promoted the expression of flowering-related genes and resulted in earlier flowering in *A. thaliana*. The *A. thaliana* plants of *JmLFY*-transformed and *JmLFY*-transformed *lfy* mutants appeared shorter leaves, longer fruit pods, and fewer cauline leaves than those of wild-type and the *lfy* mutants plants, respectively. In addition, some secondary branches in the transgenic plants converted into inflorescences, which indicated that the overexpression of *JmLFY* promoted the transition from vegetative growth to reproductive growth, and compensate the phenotypic defects of *lfy* mutant partially. The results provides a scientific reference for formulating reasonable genetic improvement strategies such as shortening childhood, improving yield and quality, and breeding desirable varieties, which have important guiding significance in production.

## INTRODUCTION

Flowering is an important process in the life cycle of higher plants. The developmental processes of flowers are divided into flowering decisions, flower initiation, and flower organ development (*Zhang & Liu, 2003*). During plant flower induction, the external environment and genetic factors together form a complex regulatory network (*Zeng et al., 2018*; *Hassankhah et al., 2018*). The model plant *Arabidopsis thaliana* controls flowering through the interaction of temperature, vernalization, photoperiod, gibberellin, and autonomous pathways (*Martina, Nadine & Christian, 2015*; *Jian et al., 2019*; *Jin et al., 2019*).

Corresponding author
Lijie Zhang, Zlj330@syau.edu.cn

In the flowering regulatory network of plants, the *LEAFY* (*LFY*) gene is a key transcription factor in the flowering signaling pathway, which is expressed in the early stage of flower primordia initiation. The *LFY* gene determines the floral meristem properties of the flower primordia by activating the expression of floral meristem characteristic genes and floral organ characteristic genes, such as *AP1* and *CAL*. Therefore, the *LFY* gene is an important floral meristem characteristic gene (*Alvarez-Buylla, Garcia-Ponce & Garay-Arroyo, 2006*; *He et al., 2018*). *LFY* and other genes related to flower formation together form a flower development regulatory network and jointly control plant flower formation. The *AtAP1* gene is involved in the formation of flower meristem, which is a characteristic gene of flower organ and plays an important role in the process of flower formation. *AP1* and *LFY* genes are positively regulated by each other. *SOC1* gene is the upstream gene of *LFY* gene, which can integrate multiple flowering pathways; the *FT* gene is also the upstream gene of *LFY* gene and the target gene of CO protein; the *PI* gene is the downstream gene of the *LFY* gene, and the *LFY* gene promotes the expression of *PI* gene (*Zhao et al., 2020*).

The *LFY* gene plays an important role not only for controlling the transition from inflorescence meristem to floral meristem but also for regulating flowering time in *A. thaliana* (*Weigel et al., 1993*; *Weigel & Nilsson, 1995*; *He, Wang & Zhang, 2011*; *He et al., 2017*). When the *LFY* gene was transferred into wild-type *A. thaliana*, it was found that all lateral shoots were transformed into flowers (*Blazquez et al., 1997*). In a functional study of *LFY* genes in plants such as *Populus simonii*, *Nicotiana tabacum*, and *Oryza sativa*, it was found that transfection and overexpression of *LFY* gene could lead to earlier flowering (*He et al., 2000*; *Ahearn et al., 2001*; *Peña et al., 2001*). The activity of the *LFY* gene is conserved even in distantly related species (*Shao et al., 1999*). Therefore, *LFY* is not only a key regulatory gene in the plant flowering pathway but also a key regulator of downstream floral meristem and floral organ-determining genes (*Feng, Li & Wang, 2016*). Although the *LFY* gene has been isolated and cloned in many plants, functional studies have shown that it plays an important role in the reproductive and growth processes of plants. However, no studies on the *LFY* homologous gene in *J. mandshurica* have been conducted.

*Juglans mandshurica* is a dichogamous and monoecious plant with two main mating types, protandrous and protogynous. The asynchronous development and the imbalanced ratios of the female and male flowers affect the efficiency of pollination and fruit setting (*Zhang et al., 2019*). For the walnut species, the fruit production is also limited by late-spring frost in many countries (*Hassankhah et al., 2017*), thus the late-leafing and early-harvesting varieties have been studied in *Juglans regia* by targeted hybridization, and a significant variation between seedlings in terms of leafing date (45 days) were observed, which indicated that it is possible to get late leafing and early-harvesting genotypes with desirable nut traits (*Hassankhah et al., 2017*; *Fallah et al., 2022*). Currently, there are few studies focus on the development of male and female flower buds and the asynchrony of flowering period. In *J. regia*, the flowering pattern was altered by spraying GA$_3$ (the best concentration is 100 mg/L), which increased the total number of flowers, the male flowers, and male: female flower ratio significantly (*Hassankhah et al., 2018*). However, the

molecular mechanism of flower formation in the process of male and female differentiation has not been fully analyzed.

Therefore, based on the successful cloning of the *JmLFY* gene in the early stage of the experiment, a plant expression vector of the *JmLFY* gene of *J. mandshurica* was successfully constructed and transformed with *Agrobacterium* to obtain transgenic *A. thaliana* plants, and the T0–T3 generations of the transgenic plants were verified and analyzed for the function of the *JmLFY* gene. This study lays a foundation for the exploration of the molecular mechanism of the flowering of *J. mandshurica*, and provides a scientific basis for the shortening of the juvenile period, early flowering, yield and quality improvement of *J. mandshurica*, selection of improved varieties, and rational formulation of genetic improvement strategies.

# MATERIALS AND METHODS

## Plant material
The test materials were obtained from the *JmLFY* gene strain stored in the Forest Genetics and Breeding Laboratory of Shenyang Agricultural University and stored at −80 °C; wild-type *A. thaliana* seeds (Columbia type) and *lfy* mutant seeds were purchased from TAIR website (https://www.arabidopsis.org/index.jsp).

## Experimental reagents
BM Seamless Cloning Kit (Biomed, Beijing, China), DH5 α and Agrobacterium GV3101 (Biomed, Beijing, China), small plasmid extraction kit (TIANGEN, Sichuan, China), Gel recovery, DL2000 Marker, High fidelity enzyme and LA Taq (TaKaRa, Beijing, China), X-gal and IPTG (Real Times, Shenzhen, China), Bis, Tris Hcl, SDS, TEMED, ammonium persulfate, ammonium persulfate, kana, sucrose, and agar powder (Sinopharm, Beijing, China), and LB medium (ShengGong, Liaoning, China).

## Acquisition of the *JmLFY* gene of *J. mandshurica*
The plasmid was extracted from the *JmLFY* cDNA bacterial solution of *J. mandshurica* stored at −80 °C, and the full-length PCR gene was amplified using high-fidelity enzymes and specific primers, the primer sequences are shown in Table 1. The PCR products were recovered by cutting the gel using a gel recovery kit and then were sent to Goldwisdom for sequencing.

## Construction of *JmLFY* gene expression vector
The plasmid pBI121 was double-digested with XbaI and SmaI, and the gel-recovered product was ligated into the pBI121 vector. The recombinant plasmid pBI121-*JmLFY* was transformed into DH5α cells and detected by PCR. The positive bacterial liquid that was successfully verified by sequencing was subjected to plasmid extraction, transformed into *Agrobacterium* GV3101 competent cells, and sent to Goldwisdom for sequencing after PCR. The successfully detected bacterial solution was used to prepare an infection solution.

**Table 1 Primer sequences used in the construction and functional verification of *Juglans mandshurica*.**

| Primer | Sequence (5′-3′) | Use |
|---|---|---|
| *LFY*-F2 | CACGGGGGACTCTAGAATGGATCCCGACCCCTTTACTG | *JmLFY vector construction* |
| *LFY*-R2 | AGGGACTGACCACCCGGGTAGAGGGGCATGTGATCACCC | *JmLFY vector construction* |
| LP | GAGGAGGAAGTGGTTACTGGG | *lfy mutant validation* |
| RP | AAATGCCTACCAAAATTATAACCG′ | *lfy mutant validation* |
| LBb1.3 | ATTTTGCCGATTTCGGAAC | *lfy mutant validation* |
| *JmLFY*-F2 | AGCACCCTTTCATTGTAACGGA | *JmLFY expression analysis* |
| *JmLFY*-R2 | GTGCTGCTATGGCGACCAAAG | *JmLFY expression analysis* |
| *Actin*-F | ATGCCCAGAAGTCTTGTTCC | *JmLFY expression analysis* |
| *Actin*-R | TGCTCATACGGTCAGCGATA | *JmLFY expression analysis* |
| *AtAP1*-F | GCTCTTAAGGCACATCCGCAC | *JmLFY functional verification* |
| *AtAP1*-R | GCAGAGGGGGAGGCATATTG | *JmLFY functional verification* |
| *AtFT*-F | ATGCCCAGAAGTCTTGTTCC | *JmLFY functional verification* |
| *AtFT*-R | TGCTCATACGGTCAGCGATA | *JmLFY functional verification* |
| *AtSOC1*-F | AGCAGAGTTGTTGGAGACGTT | *JmLFY functional verification* |
| *AtSOC1*-R | AGGTGAGGGTTGCTAGGACT | *JmLFY functional verification* |
| AtPI-F | TACAACTGGAGCTCAGGCAT | *JmLFY functional verification* |
| AtPI-R | ATTCCTCTTGCGTTGCTTG | *JmLFY functional verification* |

## Validation of *lfy* mutants

Resistance validation of *lfy* mutants of *A. thaliana* was conducted by kanamycin (KAN) screening, for which seeds of *lfy* mutants were cultured on 1/2 MS + 15 g/L sucrose + 10 g/L agar + 50 μg/mL KAN. The seeds were cultured at 4 °C for 3 d and then transferred into climatic cabinate, which maintained relative humidity of 60%, temperature of 20–22 °C, light cycle of 24 h and light intensity is 80–200 μmol/m$^2$/s.

Homozygosity validation of *lfy* mutants of *A. thaliana* was conducted by three-primer method, primers were designed using T-DNA Primer Design (http://signal.salk.edu/ tdnaprimers.2.html), the sequences of the three primers are listed in Table 1. DNA from leaves of wild type and *lfy* mutants of *A. thaliana* were extracted using TIANGEN Plant Genomic DNA Kit, which were then used for PCR. The PCR products were then detected by agarose gel electrophoresis.

## *Agrobacterium*-mediated genetic transformation of pBI121-*JmLFY* to *A. thaliana*

Genetic transformation of pBI121-*JmLFY* was conducted by inflorescence infection. Seeds of wild-type and lfy mutants of *A. thaliana* were sterilized and cultured in 1/2 MS medium, and the seedlings were selected and transplanted after 7 to 10 days of germination into the culture soil with robust and consistent growth (Fig. 1A). After bolting and flowering (Fig. 1B), the small pots were inverted, and all the inflorescences were placed upside down in the prepared infection solution, which is described in "Construction of a *JmLFY* gene expression vector" (Fig. 1C). The transformation was performed every three days and the

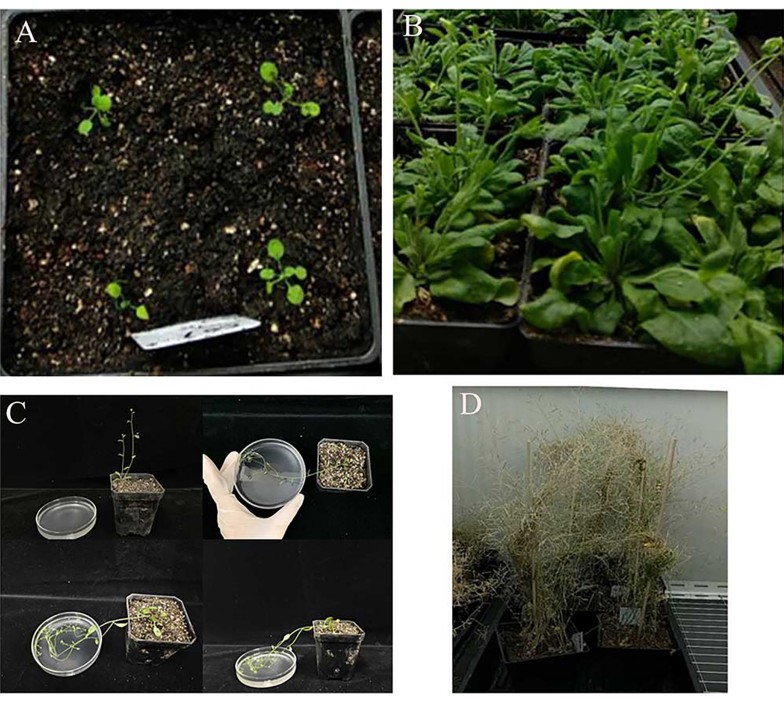

**Figure 1 Genetic transformation of *Arabidopsis thaliana*.** (A) *Arabidopsis* sowing. (B) *Arabidopsis* flowering. (C) inflorescence infection process. (D) *Arabidopsis* seed collection.

process was repeated four times, and the mature seeds (Fig. 1D) were collected for obtained regenerated plants.

## Screening of positive plants

The seeds received above were screened on 1/2 MS + KAN plates, and plants with KAN resistance were preliminarily screened. The screened plants were transferred onto the soil, and after 15 days of bolting, the DNA and RNA of the leaves were extracted using DNA and RNA extraction kits (TIANGEN, Sichuan, China), and reverse transcribed into cDNA using a cDNA synthesis kit (ShengGong, Liaoning, China), using cDNA as the template for PCR detection, and successfully detected by qRT-PCR using 2 × SG Fast qPCR master mix (Lablead).

## Overexpression of *JmLFY* gene and regulation of other genes related to flowering

The plants with high expression of *JmLFY* were harvested and cultured to the T3 generation using the same method described above. The plants with the highest expression of *JmLFY* were selected from the T3 generation plants, and the *actin* gene of *A. thaliana* was used as the internal reference gene to determine the expression of the flowering-related genes *AtAP1*, *AtFT*, *AtSOC1*, and *AtPI*. The qRT-PCR primers used are listed in Table 1.

Figure 2 Amplification of the JmLFY gene related to flowering in *Juglans mandshurica*.

## Morphological observation and statistical analysis

The selected *LFY* transgenic *A. thaliana* plants were named L1–L10, and the *LFY* transgenic *lfy* mutant plants were named C1–C10. The L-line and C-line plants with high expression of *JmLFY* were selected and cultured simultaneously with the wild-type and *lfy* mutants, and the bolting, flowering, and fruit pod formation times of the four lines were observed and recorded. The number of rosette leaves, plant height, and leaf shape and size were recorded and statistical analyses were performed using SPSS and Excel.

# RESULTS

## Acquisition of the *JmLFY* gene related to the flowering of *J. mandshurica*

The plasmid was extracted from the *JmLFY* cDNA bacterial solution of *J. mandshurica* stored at −80 °C in our laboratory. Using the *JmLFY* gene plasmid as the template, the upstream and downstream primers of the *JmLFY* gene were designed according to the obtained CDS sequence of the *JmLFY* gene, and the full-length gene was amplified by PCR. The PCR amplification products were sent to Goldwisdom for sequencing, and the sequencing results showed that the *JmLFY* target gene was successfully obtained from the cDNA-glycerol bacteria. The *JmLFY* sequence is shown in Fig. 2.

## Construction of the pBI121-*JmLFY* plant expression vector

The PCR-amplified product of the *JmLFY* gene was cut into a gel and ligated with the pBI121 vector. Sequencing and comparison showed that the homology between the two

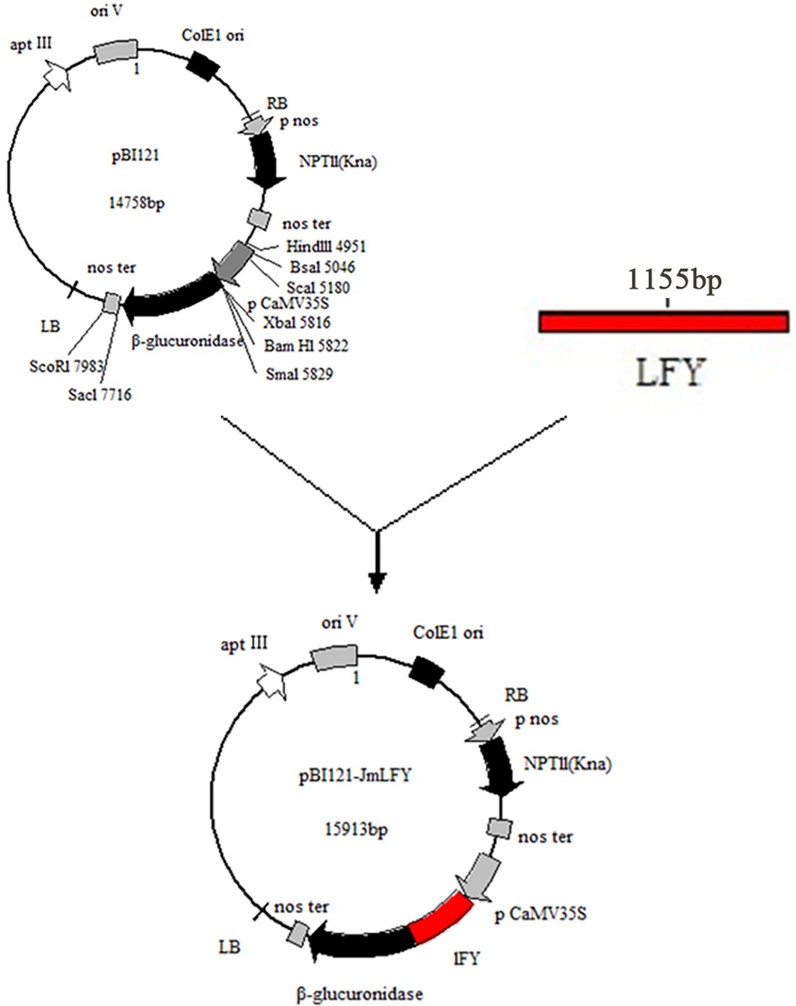

**Figure 3 Construction of plant expression vector pBI121-*JmLFY*.**

cDNA sequences reached 99.91%. The resulting plant expression vector was named pBI121-*JmLFY* and mapped (Fig. 3).

### *lfy* mutant validation

The three-primer method was used to verify the homozygosity of *A. thaliana lfy* mutants, and a 1/2 MS KAN plate was used to screen the *A. thaliana lfy* mutants for resistance. As shown in Fig. 4, the length of the wild-type band was 900 bp and that of the *lfy* mutant *A. thaliana* was approximately 800 bp. It can be concluded that the *lfy* mutant is homozygous for *Arabidopsis*, which is in line with the expected results (Fig. 5). After screening on a 1/2 MS KAN plate, it was found that the *A. thaliana lfy* mutant had yellow leaves, could not grow roots and true leaves, and could not survive for a long time (Fig. 6), indicating that it was no KAN resistant and could be used for subsequent transformation validation experiments.

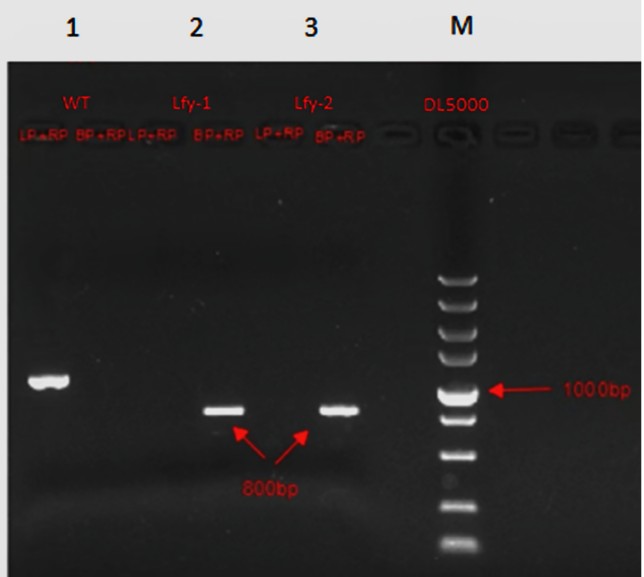

**Figure 4 Homozygous verification of *Arabidopsis*.**

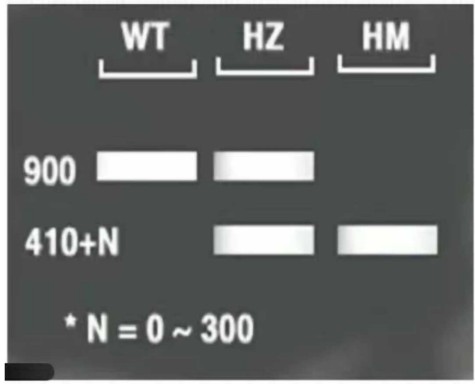

**Figure 5 Expected results for *Arabidopsis* 1–3: PCR products; M: DL5000 marker; WT: wild *Arabidosis*; HZ: heterozygote; and HM homozygous verification of *Arabidopsis*.**

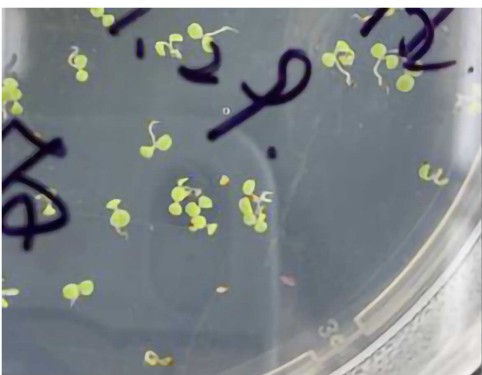

**Figure 6 Verification of kana resistance of *lfy* mutant.**

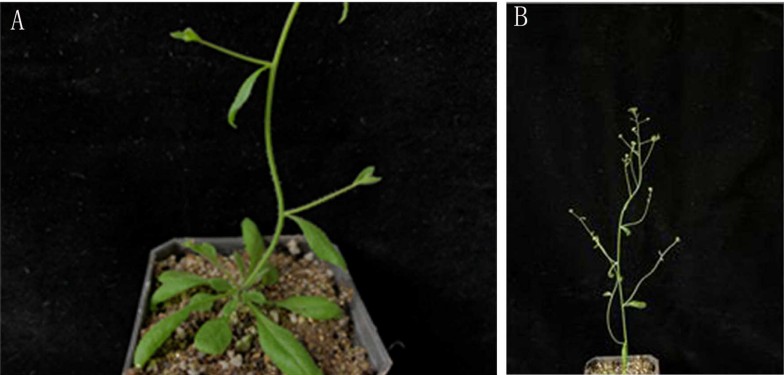

**Figure 7 (A–B)** *lfy* mutant phenotype with secondary branch.

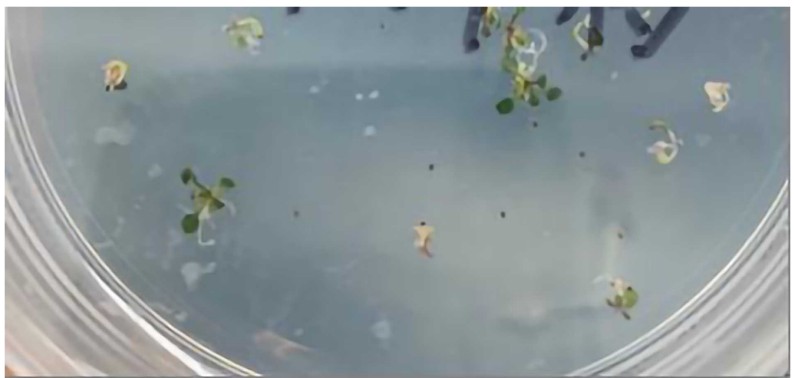

**Figure 8** Positive and false positive plants screened from the T0 generation.

Phenotypic observation showed that the inflorescence of the *lfy* mutant was transformed into secondary branches, with absent petals and incomplete flower development (Fig. 7), while the *lfy* mutant plants were highly sterile (*Zhang et al., 2008*).

## Screening of positive plants

Wild-type and *lfy* mutant *A. thaliana* transformed into the pBI121-*JmLFY* vector were screened for KAN plate resistance. The positive plants grew true leaves and took root on the KAN plate, while false positives did not (Fig. 8). At approximately 20 days, they were transferred to the prepared matrix soil. Five seedlings with the best growth were selected, and the selected *JmLFY* transgenic wild-type *A. thaliana* were named L-line plants while the selected *lfy* mutants *A. thaliana* were named C-line plants.

After flowering, the leaves were collected from the preliminarily screened lines, and the extracted RNA was reverse-transcribed into cDNA for positive PCR detection. The test results are shown in Fig. 9. L1, L2, L4, L5 and C1, C3, and C4 were found to be positive. The transformation rates of these two strains were not the same, which may be related to the quality of the recombinant plant expression vector and the operation method.

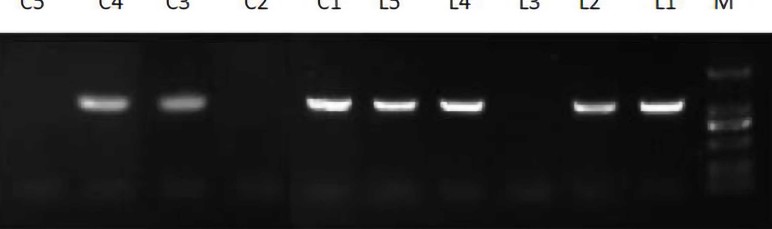

**Figure 9 Expression of *JmLFY* transgenic plants.**  

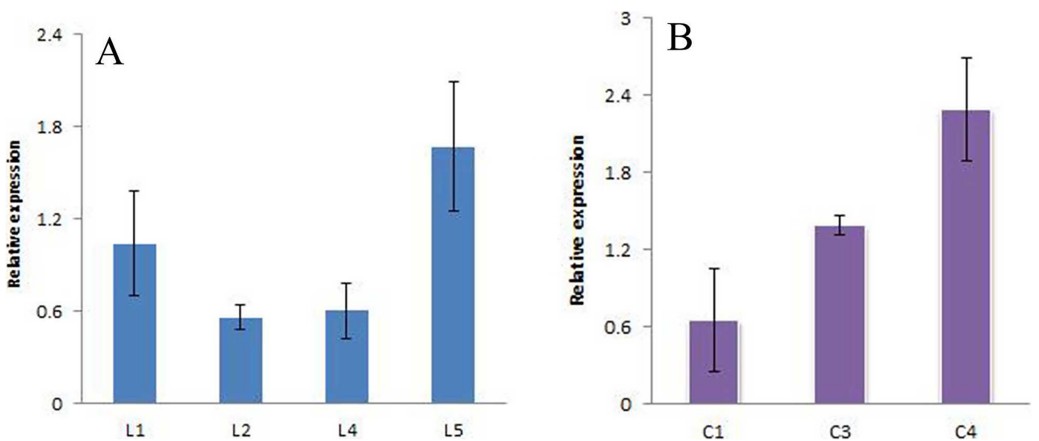

**Figure 10 Expression of *JmLFY* gene in different parts.** (A) *JmLFY* transgenic plant. (B) *JmLFY* transgenic plants with *lfy* mutant.  

The positive seedlings screened above were subjected to qRT-PCR analysis, and *A. thaliana Actin* was used as the internal reference gene to analyze the expression of the *JmLFY* gene in different lines (Fig. 10 and Supplemental File (raw data)). The C4 expression levels were found to be higher. The four harvested strains were continuously cultivated to the T3 generation, and C23, C28, and C7, which had the highest *JmLFY* gene expression, were screened for further verification.

## Regulation of *JmLFY* gene on other flowering-related genes

To further explore the regulation of other flowering-related genes in *A. thaliana* by the *JmLFY* gene of *J. mandshurica*, Primer-BLAST was used to design quantitative primers, and fluorescence quantitative analysis was used to analyze the expression of each flower-related gene in the *JmLFY* transgene, *JmLFY*-transfected *lfy* mutant, and wild type. Differences in foreign gene expression were found in different transgenic lines (Fig. 11 and Supplemental File (raw data)).

Compared to the wild type, the expression levels of all the genes related to flower formation in *Arabidopsis* plants transformed with the *JmLFY* gene changed, although the changes were different. The expression level of the *AtAP1* gene was significantly increased in L23 and C7 plants, but did not change significantly in the L28 line (Fig. 11A). The expression level of *AtSOC1* was significantly increased in L23 and C7 plants. The expression level in the L28 line was also increased, and was much higher than in

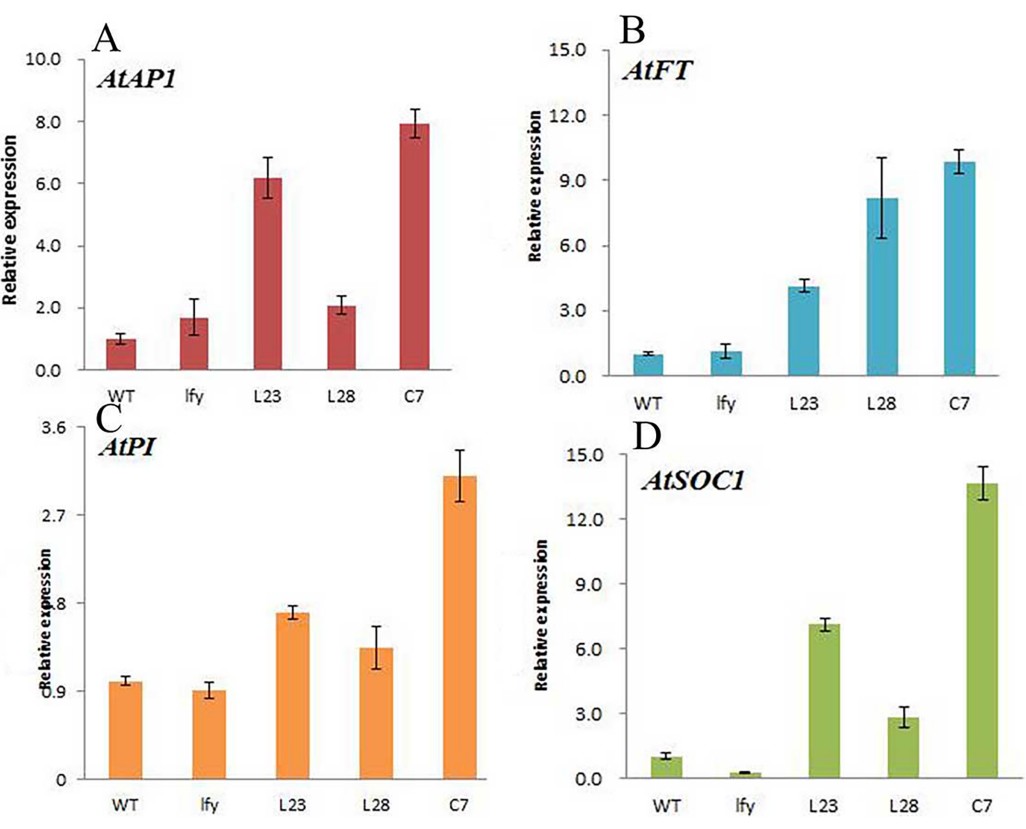

**Figure 11 Expression of different flowering genes in *JmLFY* transgenic plants.** (A) AtAP1 gene. (B) AtSOC1 gene. (C) AtPI gene. (D) AtFT gene.

**Table 2 Phenotypic data of different strains.**

| Strain | Number of rosette leaves | Flower bud formation time | Flowering time | Plant height | Number of secondary branches | Number of stem leaves |
|---|---|---|---|---|---|---|
| WT | $13.8 \pm 0.84^c$ | $31.4 \pm 1.14^c$ | $37.4 \pm 0.55^c$ | $25.92 \pm 1.12^b$ | $4.2 \pm 0.84^c$ | $5.4 \pm 1.14^c$ |
| *lfy* | $23.4 \pm 1.14^a$ | $41.6 \pm 1.14^a$ | $49.8 \pm 0.84^a$ | $29.5 \pm 0.46^a$ | $12.4 \pm 1.14^a$ | $16.8 \pm 0.84^a$ |
| L23 | $10.2 \pm 0.44^d$ | $22.8 \pm 0.84^d$ | $28.8 \pm 0.84^d$ | $25.98 \pm 1.33^b$ | $4 \pm 0.71^c$ | $4.4 \pm 1.14^{cd}$ |
| L28 | $9.6 \pm 0.89^d$ | $23.2 \pm 0.44^d$ | $29.2 \pm 0.84^d$ | $26.46 \pm 0.48^b$ | $4.42 \pm 0.84^c$ | $4 \pm 0.71^d$ |
| C7 | $18.8 \pm 0.84^b$ | $37.4 \pm 0.55^b$ | $41.2 \pm 0.84^b$ | $25.7 \pm 0.64^b$ | $7.6 \pm 0.89^b$ | $11.4 \pm 0.55^b$ |

**Note:**
The values are the means ± standard errors. Different letters in the same column indicate significant differences at $P = 0.05$ level.

wild-type *A. thaliana* (Fig. 11B). The expression of *AtFT* gene was significantly increased in the L23, L28, and C7 (Fig. 11D); the expression of *AtPI* gene in L23 and L28 lines did not change significantly, although the expression level was significantly increased in the C7 line (Fig. 11C).

The flower bud formation time, flowering time, number of rosette leaves, plant height, and number of secondary branches in the T3 generation lines are shown in Table 2 and uploaded as a Supplemental File (raw data). The phenotypes of the four lines were then compared (Fig. 12). The results showed that the flowering time of the transgenic *JmLFY*

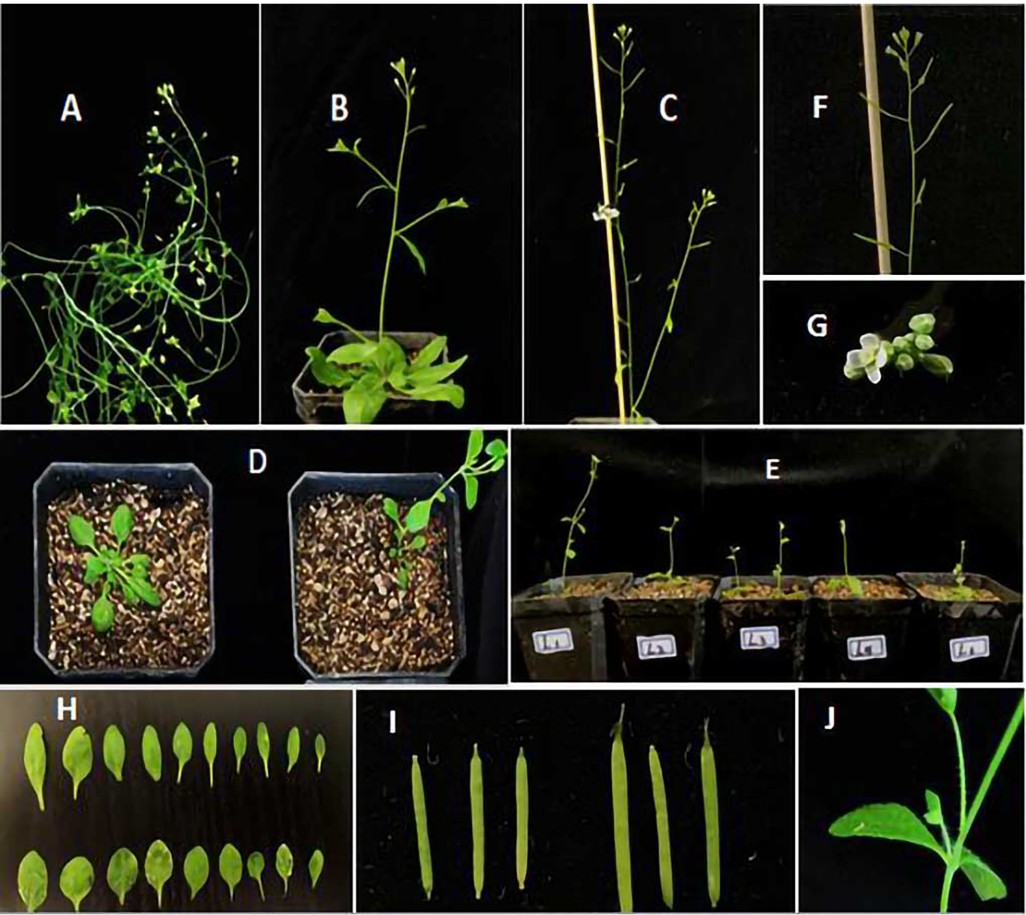

**Figure 12 Phenotypic study of overexpression of *JmLFY* gene in *Arabidopsis*.** (A) *lfy* mutant; (B) WT; (C) L23; (D) left: WT, right: L23; (E) transgenic lines flowering ahead of time; (F) transformation from secondary branch to single flower L28 (G) C7 flower; (H) up: WT down: L28; (I) WT (left) L28 (right); (J) C7 stem leaf.

plants was significantly earlier than that of the wild type; approximately eight days earlier, the rosette leaves of the L23 and L28 plants were approximately four times lower than those of the wild type, and the plant height, number of secondary branches, and number of cauline leaves were higher than those of the wild type. Compared to the *lfy* mutant, there was no obvious change, and the flowering time of the transfected *lfy* mutant was still later than that of the wild-type *A. thaliana*; however, it was also significantly earlier than that of the *lfy* mutant by approximately 7 days. The number of rosette leaves in the C7 plant was higher than that of the *lfy* mutant. Approximately five fewer leaves, four fewer secondary branches, and five fewer cauline leaves (Table 2); transgenic plants have shorter leaves and longer pods than the wild type (Figs. 12H and 12I), and all shoots were transformed into single flowers several times (Fig. 12F), indicating that *JmLFY*, as a characteristic gene of floral meristem, can control the transition from vegetative growth to reproductive growth. *lfy* mutants are sterile plants, and the flowers remained intact after the transfer of the *JmLFY* gene (Fig. 12G). Most of the flowers regained their normal morphology, produced more seeds than the *lfy* mutant, and produced fewer rosette leaves, secondary shoots, and

cauline leaves than the *lfy* mutant (Fig. 12J). Some secondary branches were also transformed into single flowers, and *JmLFY* also transformed secondary branches into inflorescences. In addition, *JmLFY* regulates the expression of other flowering-related genes in *A. thaliana* and promotes the expression of these genes, indicating that the function of *JmLFY* may also be related to the regulation of these genes. These results illustrate the important role of *JmLFY* in flower formation.

## DISCUSSION

The transition from vegetative to reproductive growth is an important process in the plant life cycle, and many physiological and metabolic changes and gene regulation mechanisms are involved in this process (*Saquib & Li, 2020*; *Dong et al., 2021*). These gene regulatory networks do not exist independently but interact and regulate each other to jointly regulate the flowering process of plants (*Lu et al., 2020*; *Kalve et al., 2020*). The *LFY* homologous gene regulates the formation of floral meristems and organs and is critical for flowering (*Bosl et al., 2004*). As a transcription factor, *LFY* regulates the transition of plants from vegetative growth to reproductive growth and is mainly expressed in the apical meristem. Under the control of the 35S promoter, *LFY* overexpression promotes early flowering and transforms secondary shoots into single flowers (*Zhang et al., 2008*). *Lfy* mutants are defective in flowering and inflorescence formation (*Mori et al., 2017*; *Ma et al., 2020*). When *GhLFY* is overexpressed in *A. thaliana lfy*-5 mutants, lateral shoot secondary buds are transformed into individual flowers and the wild-type flower phenotype is restored (*Tang et al., 2016*). In this study, the secondary shoots of the transgenic *A. thaliana* were transformed into single flowers, the mutant plants of the transgenic *JmLFY* gene exhibited normal petals, the phenotype of the mutant plants was restored to normal, and the seeds produced are more and better quality than the *lfy* mutant. These results are similar to those of mutant *lfy* overexpressing the *JcLFY* gene of *Jatropha curcas* (*Tang et al., 2016*) and the *EgLFY* gene of *Eucalyptus grandis* (*Dornelas & Rodriguez, 2006*), indicating that the *JmLFY* gene can complement the late-flowering phenotype of *lfy* mutants and promote the transition from vegetative growth to reproductive growth.

Plant flower bud differentiation is a complex physiological process regulated by multiple genes (*Zhang et al., 2019*). In *J. regia*, expression of *FT* gene activated downstream floral meristem identity genes including *SOC1*, and *LFY* which consequently led to release bud dormancy as well as flower anthesis and induction (*Hassankhah et al., 2020*). As a master regulator of flowering and floral gene networks, *LFY* activates downstream floral meristem recognition genes (*AP1* and *CAL*) (*Zou et al., 2014*). *AP1* is a floral meristem signature gene and a direct target of *LFY* (*Ma et al., 2020*). *AP1* mRNA gradually accumulates after *LFY* expression (*Haughn, 1991*). In addition, *LFY* also promotes the expression of floral organ-specific recognition genes, such as *PI*, *AG*, and *AP3*, as well as the E functional genes *SEP1*, *SEP2*, and *SEP3*. In this study, compared with the wild type, the expression of *AtAP1* gene was significantly increased in L23 and C7 plants, but did not significantly change in the L28 line; the expression of *AtSOC1* gene was significantly increased in L23 and C7 plants, and the expression level of the L28 line was also higher than that of wild-type *A. thaliana*. The expression level of the *AtFT* gene in L23 and L28 was significantly

increased; and the expression level of *AtPI* gene in L23 and L28 lines showed no significant change; however, the expression level was significantly increased in the C7 line. This indicated that the three genes *AtAP1*, *AtSOC1*, and *AtPI* may contribute to the normal flower morphology of C7.

Transferring *JcLFY* into *Jatropha curcas* caused transgenic plants to bloom seven months earlier (*Tang et al., 2016*). *Peña et al. (2001)* transformed the *AtLFY* and *AtAP1* genes into *Citrus sinense* and *Poncirus trifoliata*, and found that transformed plants flowered 3–5 years earlier than untransformed plants (*Pineiro, 1998*; *He, Wang & Zhang, 2011*; *Ma et al., 2019*). In the present study, the transgenic *A. thaliana* L23 and L28 lines flowered six days earlier than the wild-type plants, and the C7 line flowered eight days earlier than the corresponding mutant *lfy*. These results suggest that *JmLFY* promotes early flowering in *A. thaliana*, shortens the vegetative phase, and complements the delayed flowering of the mutant *lfy*. In addition, the L23 and L28 lines had fewer rosette and cauline leaves than the wild type. And also, the number of clump leaves, cauline leaves, and branches of the transgenic line C7 were higher than those of the wild-type plants but lower than those of the *lfy* mutant. The results obtained in this study are in agreement with those obtained in *A. thaliana* (*Weigel & Nilsson, 1995*), New Zealand radiata pine (*Vazquez et al., 2007*), rapeseed (*Roy, Saxena & Bhalla, 2009*) and *Jatropha curcas*. The phenotypes of plants overexpressing the *LFY* gene in *Tang et al. (2016)* were similar. These results confirm that *JmLFY* can inhibit the vegetative growth of plants, partially complement the phenotypic defects of *lfy* mutants, and promote the earlier flowering of plants. And in the future research, the Southern Blot method should be considered to better confirm the transgenic plants.

## CONCLUSIONS

In this study, the plant expression vector pBI121-*JmLFY* was successfully constructed using *JmLFY* cDNA bacterial solution. The three-primer method and KAN resistance screening confirmed that the *lfy* mutant was homozygous and did not exhibit KAN resistance. Phenotypic observations showed that the inflorescence of *lfy* mutant *Arabidopsis* was transformed into secondary branches, with absent petals and incomplete flower development. The *lfy* mutant plants were highly sterile and produced few seeds. pBI121-*JmLFY* was transformed into wild-type and *lfy* mutants to obtain regenerated plants, and the *JmLFY*-transgenic *Arabidopsis* and *JmLFY*-transgenic *Arabidopsis* were screened for positive seedlings using KAN plates. Expression of the *JmLFY* gene was screened and plants with the highest levels of expression were those from the L23, L28, and C7 lines. Validation of the T3 generation plants showed that *JmLFY* also regulated the expression of other flowering-related genes in *A. thaliana* and promoted the expression of these genes. The *JmLFY* gene can partially compensate for the phenotypic defects of *lfy* mutants, and promote transition from vegetative to reproductive growth and early flowering.

### Funding

This work was supported by the Scientific Research Project of Liaoning Education Department (LSNZD201905), the Key Project of Liaoning Natural Science Foundation of China (No. 20170540801), the 13th 5-year National Key Research and Development Plan Project (No. 2017YFD0600600) and the Liaoning Province College Students Innovation and Entrepreneurship Training Program (2021-114). The funders had no role in study design, data collection and analysis, decision to publish, or preparation of the manuscript.

### Grant Disclosures

The following grant information was disclosed by the authors:
Scientific Research Project of Liaoning Education Department: LSNZD201905.
Key Project of Liaoning Natural Science Foundation of China: 20170540801.
13th 5-year National Key Research and Development Plan Project: 2017YFD0600600.
Liaoning Province College Students Innovation and Entrepreneurship Training Program: 2021-114.

### Competing Interests

The authors declare that they have no competing interests.

### Author Contributions

- Jiayou Cai conceived and designed the experiments, performed the experiments, analyzed the data, authored or reviewed drafts of the article, and approved the final draft.
- Ruoxue Jia conceived and designed the experiments, analyzed the data, prepared figures and/or tables, and approved the final draft.
- Ying Jiang conceived and designed the experiments, performed the experiments, analyzed the data, prepared figures and/or tables, authored or reviewed drafts of the article, and approved the final draft.
- Jingqi Fu conceived and designed the experiments, analyzed the data, prepared figures and/or tables, and approved the final draft.
- Tianyi Dong performed the experiments, analyzed the data, prepared figures and/or tables, authored or reviewed drafts of the article, and approved the final draft.
- Jifeng Deng conceived and designed the experiments, performed the experiments, analyzed the data, prepared figures and/or tables, and approved the final draft.
- Lijie Zhang conceived and designed the experiments, performed the experiments, analyzed the data, authored or reviewed drafts of the article, and approved the final draft.

### Data Availability

The raw data is available in the Supplemental File.

## Supplemental Information

Supplemental information for this article can be found online at http://dx.doi.org/10.7717/peerj.14938#supplemental-information.

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
