# Peer review of "Functional verification of the JmLFY gene associated with the flowering of Juglans mandshurica Maxim"

_PeerJ, doi:10.7717/peerj.14938_

## Round 0.1 · original submission · Major Revisions

Follow the reviewers comments.

Reviewer 3 has requested that you cite specific references. You may add them if you believe they are especially relevant. However, I do not expect you to include these citations, and if you do not include them, this will not influence my decision.

·

Basic reporting

English revision and citation style needed to be improved

Experimental design

Methodology is defined and organized well

Validity of the findings

Manuscript have novel findings, figures quality is publishable

Additional comments

N/A

·

Basic reporting

no comment

Experimental design

no comment

Validity of the findings

no comment

Additional comments

In this manuscript, authors identified the full-length sequence of the JmLFY gene, and constructed
a pBI121-JmLFY plant expression vector. And using wild-type Arabidopsis and lfy mutants to explore the molecular mechanism of the development of Juglans mandshurica hermaphroditic flowers. I found this topic interesting but I have many concerns related to the manuscript. In my opinion, introduction is well-written. However, part of the methods section requires detailed information. There are still good rooms to improve the quality of the manuscript. I recommend the authors to have all my comments addressed and revise the manuscript.
1.Line 99, TAIR's URL should be added..
2.Line 121-122, The method should be more detailed. And when the word first appears in the manuscript, the full name should be displayed, such as KAN.
3.Line 185-188, make sure you present only results in the Result Section. This sentence is more suitable for the method part. Please check the entire manuscript for similar situations.
4.I suggest putting the sequence information of Figure 1 in the Supplemental files.
5.The resolution of Figure 2, 3 is not high enough, it is blurry.
6.The Figures should be further improved in the manuscript, I think these Figures cannot be published. In addition, the legends of Figures need to be more accurate.
7.I also highly recommend the professional language editing of the manuscript before resubmisson by American Journal Experts, or by any other similar companies, such as Line 109, “in our laboratory”, it's really bad.

Reviewer 3 ·

Basic reporting

.

Experimental design

.

Validity of the findings

.

Additional comments

I reviewed the paper titled: Functional verification of the JmLFY gene associated with the flowering of Juglans mandshurica Maxim.
It is a good research work with practical results.
Any research on flowering fruit trees and shortening the juvenile period is valuable.
But the physiology of fruit trees is a bit more complicated and some different than that of annual plants.
There are many candidate genes for induction of early fertility and inflorescence development. Why was this gene (LFY) chosen?
You could use the Southern Blot method to confirm the transgenic plants better.
The photos were not visible in the file that was in my hand.
You should enrich the introduction and discussion using the relevant papers. For example it is recommended to address the following papers:
Hassankhah A, Vahdati K, Rahemi M, Sarikhani Khorami S (2017) Persian walnut phenology: Effect of chilling and heat requirements on budbreak and flowering date. International Journal of Horticultural Science and Technology 4(2):259-71.
Hassankhah A, Rahemi M, Mozafari MR, Vahdati K (2018) Flower development in walnut: altering the flowering pattern by gibberellic acid application. Notulae Botanicae Horti Agrobotanici Cluj-Napoca. 46(2):700-706.
Hassankhah A, Rahemi M, Ramshini H, Sarikhani S, Vahdati K. (2020). Flowering in Persian walnut: patterns of gene expression during flower development. BMC Plant Biology, 20(1): 1-10.
Fallah M, Vahdati K, Hasani D, Rasouli M, Sarikhani S (2022) Breeding of Persian walnut: Aiming to introduce late-leafing and early-harvesting varieties by targeted hybridization. Scientia Horticulturae, 295.
Hassani D, Sarikhani S, Dastjerdi R, Mahmoudi R, Soleimani A, Vahdati K (2020). Situation and recent trends on cultivation and breeding of Persian walnut in Iran. Scientia Horticulturae, 270: 109369.

---

## Round 0.2 · accepted · Accept

The article has been revised as per reviewers comments/suggestions therefore, I am recommending acceptance.

·

Basic reporting

Authors have incorporated all the pointed changes and now the manuscript is acceptable in its current form.

Experimental design

Experimental design is well organized

Validity of the findings

No comments

·

Basic reporting

no comment

Experimental design

no comment

Validity of the findings

no comment

Additional comments

Based on my suggestion in the first review, the authors have addressed my concerns appropriately. So, the manuscript can be accepted for publication in PeerJ.